# Nitrite Determination in Environmental Water Samples Using Microchip Electrophoresis Coupled with Amperometric Detection

**DOI:** 10.3390/mi13101736

**Published:** 2022-10-14

**Authors:** Simone Bernardino Lucas, Lucas Mattos Duarte, Kariolanda Cristina Andrade Rezende, Wendell Karlos Tomazelli Coltro

**Affiliations:** 1Instituto de Química, Universidade Federal de Goiás, Goiânia 74690-900, GO, Brazil; 2Instituto de Química, Departamento de Química Analítica, Universidade Federal Fluminense, Niterói 24020-141, RJ, Brazil; 3Instituto Nacional de Ciência e Tecnologia de Bioanalítica (INCTBio), Campinas 13083-861, SP, Brazil

**Keywords:** chip-based electrophoresis, environmental monitoring, miniaturized systems, nitrogen cycle, portable instrumentation

## Abstract

Nitrite is considered an important target analyte for environmental monitoring. In water resources, nitrite is the result of the nitrogen cycle and the leaching processes of pesticides based on nitrogenous compounds. A high concentration of nitrite can be associated with intoxication processes and metabolic disorders in humans. The present study describes the development of a portable analytical methodology based on microchip electrophoresis coupled with amperometric detection for the determination of nitrite in environmental water samples. Electrophoretic and detection conditions were optimized, and the best separations were achieved within 60 s by employing a mixture of 30 mmol L^−1^ lactic acid and 15 mmol L^−1^ histidine (pH = 3.8) as a running buffer applying 0.7 V to the working electrode (*versus* Pt) for amperometric measurements. The developed methodology revealed a satisfactory linear behavior in the concentration range between 20 and 80 μmolL^−1^ (R^2^ = 0.999) with a limit of detection of 1.3 μmolL^−1^. The nitrite concentration was determined in five water samples and the achieved values ranged from (28.7 ± 1.6) to (67.1 ± 0.5) µmol L^−1^. The data showed that using the proposed methodology revealed satisfactory recovery values (83.5–103.8%) and is in good agreement with the reference technique. Due to its low sample consumption, portability potential, high analytical frequency, and instrumental simplicity, the developed methodology may be considered a promising strategy to monitor and quantitatively determine nitrite in environmental samples.

## 1. Introduction

Inorganic ions are widely considered as parameters for monitoring and controlling the quality of food samples [1,2], biological matrices [3,4], environmental samples [5,6], and clinical diagnoses [7,8]. Among the inorganic ions, nitrite has been studied to evaluate food quality. Cardoso and coworkers determined the presence of nitrite in sausage and ham samples, due to the use of this compound as a food preservative [9,10]. In biological matrices, nitrite is evaluated due to its relationship with the metabolic pathway of nitrous oxide and a series of other clinical diagnoses, depending on the fluid in which it is evaluated (urine, saliva, or blood) [10,11,12]. In the field of environmental analysis, the presence of nitrite in water samples comes from natural processes such as biological denitrification and acid rain, or it is due to contamination by industrial waste and other economic activities [13,14].

In recent years, fertilizers and agrochemicals based on nitrogen species have been increasingly used, and in the case of an over-application, the lixiviation process can transfer nitrite to rivers and lakes [15,16,17]. The consumption of water and food containing nitrite above the limit can cause its accumulation in the human body [18]. In high concentrations, nitrite may be associated with methemoglobinemia or “blue baby syndrome” [19], carcinogenic nitrosamines, gastric cancer, spontaneous intrauterine growth restriction, abortions, and birth defects of the central nervous system, among other diagnoses [20,21,22].

The quantification of nitrite in environmental water supply samples is an important procedure to monitor and control the increase of NO_2_^−^ in water resources and avoid possible poisoning [18,23,24]. Nitrite can be determined by spectrophotometric measurements through different methodologies including nitrosation-based [25], catalytic [26], and Griess reaction [14] assays. The latter is the most widely used due to its simplicity and the low cost per analysis; however, bench-based standard methodologies require a large volume of reagents, generate a considerable amount of waste, and may be considered time-consuming [14,25].

Other techniques have also been employed for nitrite analysis like chemiluminescence, chromatography, electrochemistry, and capillary electrophoresis (CE) [27]. Chemiluminescence provides simplicity, a wide linear range, and low cost, but it also offers poor stability and reproducibility [28,29,30,31,32]. Chromatographic methods such as gas chromatography hyphenated with mass spectrometry detection (GC-MS) and high-performance liquid chromatography (HPLC) are explored more often due to their high selectivity and the potential to perform a faster analysis [33,34,35,36]; however, they require high-cost instrumentation, sample pre-treatment, and derivatization steps [25]. Electrochemical detection has also been explored for nitrite analysis, offering low-cost analyses with miniaturization capability. In addition, the use of modified electrodes helps to potentially increase sensitivity and selectivity [37,38,39,40,41]. Lastly, conventional CE systems have gained prominence among the available separation techniques due to their efficiency, resolution, and relatively short analysis times, when compared to chromatographic separations [7].

With the aim of employing a faster alternative approach for the detection of nitrite, portable microsystems have been described in the literature, including paper-based colorimetric assays [38]. Nevertheless, paper-based devices can suffer interference from other ions, as well as limiting the possibility of using some solvents due to their interaction with the platform material [10,17,42,43]. Another alternative is the use of microchip electrophoresis (ME) devices due to their potential for portability and high separation efficiency, short analysis time, reduced sample consumption, and low waste generation [44]. In addition, ME devices can be manufactured on different materials including glass, silicon, polymers, threads, 3D printing filaments toner, and paper-based platforms on paper or polymers [45,46,47,48,49].

Freitas and coworkers used ME with capacitively coupled contactless conductivity detection (C^4^D) to monitor inorganic species, including nitrite in aquarium and river water, achieving a limit of detection (LOD) of 4.9 µmoL^−1^ [50]. In addition to conductometric detection [50,51,52], amperometric detection (AD) can be used in association with chip-based systems, because it provides better selectivity and sensitivity [53,54,55,56]. For environmental samples, selectivity is an important feature to be considered, especially because the presence of other charged species in water samples, such as chloride, nitrate, and sulfate anions, can negatively affect the separation performance when conductivity detection modes are used [57,58]. To increase the selectivity and sensitivity of amperometric detection for nitrite determination, different groups have proposed the use of simple hybrid electrodes, nanotubes, and/or nanocomposites [22,24,59,60].

In this context, the present study proposes the use of ME–AD for the quantification of nitrite in environmental water samples. The methodology was developed using a commercial and portable instrument, which comprises a potentiostat, a high voltage power supply, and a microfluidic platform to assemble ME devices with integrated electrodes. The feasibility of the proposed methodology was demonstrated through nitrite determination in water samples from two types of ecosystems, a small aquarium, and a fish breeding dam. The obtained results were compared with the standard spectrophotometric method based on the Griess reaction.

## 2. Materials and Methods

### 2.1. Reagents and Solutions

Histidine (His), sodium hydroxide (NaOH), sulphanilamide, and N-(1-naphtyl)-ethylenediamine dihydrochloride (NED) were purchased from Sigma–Aldrich (St. Louis, MO, USA). Lactic Acid (HLat) was acquired from Cromoline (Diadema, SP, Brazil), and sodium nitrite (NaNO_2_) and citric acid were supplied by NEON (Sao Paulo, SP, Brazil). Stock solutions of HLat, His, NaOH, and NaNO_2_ were prepared at concentrations of 200, 100, 100, and 10 mmol L^−1^, respectively. For the reference methodology using the Griess reaction, stock solutions of 4 mmol L^−1^ NED and 50 mmol L^−1^ sulphanilamide were used. All solutions were weekly prepared with ultrapure water (resistivity 18 MΩ cm) and filtered through nylon filters with 0.22 µm pore diameter. All experiments were performed at 23 ± 1 °C.

### 2.2. Samples

Water samples were collected from two types of ecosystems located in the city of Inhumas-GO: a small aquarium (16°22′24.9″ S 49°29′04.4″ W) and a fish breeding dam (16°20′16.8″ S 49°29′44.4″ W). Both types of samples were collected, filtered through nylon filters with 0.22 µm pore diameter, and stored in polymeric sterile tubes. No further pretreatment was necessary, and dilutions were made for quantification by the reference and the proposed methods.

### 2.3. Instrumentation

For the analytical procedures performed with amperometric detection, an HVStat system supplied by MicruX Technologies (Asturias, Spain) was used. This system consists of a portable instrument that includes a high voltage power supply integrated with a 165 × 150 × 85 mm^3^ potentiostat (Figure 1A) and a platform (Figure 1B) that has been specifically developed for use by coupling the microchips with the detection system. This system also incorporates software (MicruX Manager) for instrumentation control, data acquisition, and processing.

The electrophoresis microchip was composed of a hybrid SU-8/glass platform (38 mm long × 13 mm wide × 0.8 mm thick) (Figure 1C) containing injection and separation microchannels (50 µm width × 20 µm deep) designed in a cross-shaped arrangement. The total and effective separation microchannel lengths were 35 and 30 mm, respectively. Amperometric detection was performed using integrated Ti/Pt (50/150 nm) thin-film electrodes, where the working electrode (WE) was positioned at 20 µm from the separation channel extremity (end-channel mode). All three electrodes were spaced at 100 μm from each other.

### 2.4. Electrophoretic Procedure

The microchips were preconditioned and rinsed with 0.1 mol L^−1^ NaOH, ultrapure water, and running buffer for 30, 15, and 10 min, respectively. The procedure was performed with the aid of a simple vacuum system, at 560 mmHg. After the preconditioning stage, all the reservoirs and microchannels were filled with the running buffer solution.

The microchip was connected to the HVStat system while fixed in its holder and the baseline was stabilized. The sample was electrokinetically injected using the floating mode, where an electric potential was applied for enough time to fill the injection channel [61,62,63]. Following this, another electric potential was applied so that the sample volume at the intersection of the channels went to the detection zone. Finally, the detection potential was applied and the electropherogram was recorded. All experiments were performed at 23 ± 1 °C.

### 2.5. Reference Analytical Method

To assess the accuracy of the developed methodology, the samples were also analyzed according to the spectrophotometric procedure based on the Griess reaction, following the protocol of the Association of Official Analytical Chemists [64]. After the colorimetric reaction, measurements were carried out with a UV-Vis Spectrophotometer manufactured by FEMTO (São Paulo, SP, Brazil) at 530 nm.

## 3. Results and Discussion

### 3.1. Detection Potential Optimization

Hydrodynamic voltammograms were recorded to evaluate the best potential for nitrite detection during electrophoretic procedure. A standard solution of 200 µmolL^−1^ NO_2_^−^ was then used as a model in electrophoretic runs and detected by applying potentials between 0.5 and 1.0 V to the working electrode (see the electropherograms presented in Appendix A—available in the Appendix A). The signals were evaluated for the area, intensity, and peak width. In addition, the stability of the electric current was observed. Considering all these parameters, the potential of 0.7 V was defined as optimum and kept constant for the next steps due to providing the highest intensity and peak area, as shown in Appendix A. Furthermore, it is important to mention that the baseline current remained stable for the analysis that was applying this detection potential. For a higher potential, a noticeable increase in the background current was observed (Appendix A).

### 3.2. Optimization of the Running Buffer

According to the literature, the separation of the anionic species by ME devices is successfully performed when lactic acid is used on the running buffer [50,51]. Therefore, a buffer solution composed of lactic acid and histidine was prepared at different ratios to investigate the pH effect on nitrite determination. Th analyses were carried out in an acidic medium to favor nitrite oxidation [65,66,67,68]. For this purpose, the lactic acid concentration was fixed at 30 mmol L^−1^ and the histidine concentration ranged from 5 to 25 mmol L^−1^ (5 mmol L^−1^ increments), providing solutions with pH at 3.2, 3.6, 3.8, 4.1, and 4.5, respectively (a–e) (Figure 2A). An increase in the concentration of histidine led to an increase in the pH of the electrolyte and an improvement in peak shape. However, at 20 mmol L^−1^ of histidine, the peak area decreased, and the peak width had widened (Figure 2B), which negatively impacted the sensitivity and separation efficiency. Thus, the run buffer that was composed of 30 mmol L^−1^ lactic acid and 15 mmol L^−1^ histidine (pH = 3.8) was defined as optimum and kept constant once it also provided the lower relative standard deviation values.

### 3.3. Eletrophoretic Parameters

The sample was electrokinetically introduced into microchannels using the floating mode. To achieve the suitable injection time, a 50 µmol L^−1^ nitrite standard solution was used as model. For this purpose, injection time was varied from 1 to 20 s. As shown in Figure 3A, the presence of nitrite was observed only for injection times longer than 3 s. As expected, the increase in the injection time caused a greater volume of the sample to be injected and, consequently, a higher signal was observed [63]. For injection times longer than 10s, a noticeable peak broadening with a consequent loss of peak symmetry was observed. Based on the data presented in Figure 3B, the injection time of 10 s was then selected as ideal for the subsequent experiments.

The potentials used for electrokinetic control of solutions into microchannels were also optimized. The injection potential was varied from −300 to −1000 V, and separation from −500 to −1200 V (Figure 4A). The peak width and migration times presented similar behaviors, being inversely proportional to the application of potential. The peak area, as expected, was directly proportional to the increase in the applied potential (Figure 4B).

When applying −1200 V for the separation potential, despite obtaining the highest peak area and the shortest migration time, a slightly higher RSD value (8.6%, *n* = 3) was obtained. Therefore, −1000 V was defined for the separation potential since the difference in the area for both conditions was low, and under these conditions a lower RSD was obtained between the replicates (Figure 4B).

The effect of the difference in conductivity between the sample and the running buffer, in the detection, was also evaluated (Appendix A). To achieve this, a fortified sample and a standard solution were prepared at the same concentration (30 μmolL^−1^ of NO_2_^−^). The samples were diluted in water and a 10% *v/v* running buffer. When comparing the nitrite peak areas and intensities recorded under the two dilution procedures, it was observed that samples diluted in 10% *v/v* buffer presented a smaller standard deviation. The difference in the peak area in the fortified sample compared to that obtained in the standard solution was lower than 12.4%. This may be attributed to a biased electrokinetic injection and possibly to sample stacking. Therefore, a 10% *v/v* running buffer was used in all subsequent dilutions.

### 3.4. Method Validation

After experimental optimizations, the analytical performance of the proposed methodology was investigated by consecutive injections of a standard nitrite solution prepared at 100 μmol L^−1^. For a sequence of three sequential injections, the relative standard deviation (RSD) values for migration time, peak area, and intensity ranged from 0.1 to 2.3% (Table 1), thus suggesting satisfactory injection–to–injection repeatability in an intra-day comparison. In the same way, the nitrite analysis was also performed on three different days using two different commercial microchips of the same model (MCE-SU8-Pt001). In the inter-day comparison (*n* = 3), the RSD values for the measured parameters varied from 3.3 to 15.6%. Lastly, for an inter-chip comparison and based on the recorded electropherograms, the RSD values calculated for migration time, peak area, and intensity varied between 3.4 and 11.8%. All the data obtained are summarized in Table 1 and the variations observed may be associated with slight changes on the electroosmotic flow mobility. Nevertheless, the comparison discussed herein demonstrates the potential of SU-8/glass devices for routine analysis employing a portable instrument, which may enable its use for in-field assays.

In addition to the comparisons discussed above, the analytical performance was also investigated in terms of linear range and detectability levels. The developed method exhibited good linear behavior (R^2^ = 0.999) in the concentration range between 20 and 80 μmol L^−1^ (Area = −7.123 + 0.692 × [NO_2_^−^]; R^2^ = 0.999). The limit of detection (LOD) was calculated based on the ratio between three times the standard deviation for the blank and the angular coefficient of the analytical curve and the value was 1.3 μmol L^−1^. The LOD value achieved using the ME–AD system was compared to other reports published in the last five years involving miniaturized and conventional techniques, as summarized in Table 2. In addition to the LOD, other features such as analysis time, tested sample, employed technique, and portability ware also included.

As shown in Table 2, well-established analytical techniques have provided better LOD values. However, most of the examples exploring these techniques require bulky and costly instrumentation, which are restricted to a few research groups and are not compatible with portability, making their in-field use difficult [3,24,26,44,59,60]. Furthermore, when compared to paper-based devices [38,43], the portability of ME–AD has a noticeably lower appeal. On the other hand, ME–AD devices can be reused many times (estimated up to 1000 analyses) and, although not demonstrated in this study, they can allow selective analysis in the presence of other anionic species.

In comparison to the recent studies employing the ME–AD, our proposed methodology has provided one of the lowest LOD values that was achieved using unmodified electrodes, thus revealing attractive advantages over other methods which use modified electrodes and sample pretreatment steps [56]. Moreover, based on the examples using ME devices with portable instrumentation, the device explored in this study has offered the shortest analysis time. In view of the studies compared in Table 2, the performance obtained through the proposed method as well as the previously discussed advantages make clear its potential for in-field analysis.

### 3.5. Environmental Water Analysis and Comparison with a Reference Methodology

Environmental samples of aquarium water (A) and fish breeding dam water (D) were analyzed and the recorded electropherograms are displayed in Figure 5. Samples were diluted 50% (*v*/*v*) and then spiked with nitrite standard solution. Thus, samples labelled as A1, A2, and A3 correspond to the signal obtained for the analysis of aquarium water diluted at 50% (*v*/*v*) and fortified with 30, 40, and 60 μmol L^−1^ of NO_2_^−^, respectively. On the other hand, samples labelled as D1 refer to a sample of dam water diluted by 50% (*v*/*v*), while samples D2 and D3 are the same dam water diluted but fortified with 20 and 40 μmol L^−1^ of NO_2_^−^, respectively. All six samples (A1, A2, A3, D1, D2, and D3) were analyzed in triplicate by both the proposed and the reference methods. The aquarium samples (A) showed a lower concentration of nitrite due to the nitrogen cycle, presenting concentrations below the detection limit for both the proposed method and the reference method. Freitas and collaborators also found this nitrite profile in aquarium water samples in the first weeks (0 to 8) of operation; however, due to the formation and development of the ecosystem of a small aquarium, the conversion of species to nitrite is not very significant, and the aquarium where the samples were collected had recently been cleaned and reassembled [50].

Reference analyses were carried out using a spectrophotometric procedure based on the Griess reaction. The obtained linear regression equation related to the analytical curve was: Absorbance = 0.002 + 0.037 × [NO_2_^−^] (R^2^ = 0.999). The original dam water sample (D) showed a NO_2_^−^ concentration of 64.4 ± 0.6 μmol L^−1^. Based on this, the final concentration of NO_2_^−^ in samples D1, D2, and D3 were 32.2 ± 0.3; 52.2 ± 0.5, and 72.2 ± 0.6 μmol L^−1^, respectively. The electropherograms obtained for the analysis of each of the samples are shown in Figure 5. It was possible to observe the increase of the signal referring to the nitrite after the fortifications. There was also a difference in the profile of the signals obtained for nitrite according to the sample matrix. For aquarium samples (A1–A3) the nitrite peaks had a lower intensity and peak symmetry, resulting in wider signals. In the samples from dam water (D1–D3), the signals obtained were more intense and symmetrical.

This difference in the peak symmetry can be justified by the fact that the aquarium water was collected only one day after changing the filters. In addition, the aquarium fish population was entirely the same species, Kinguio (Carassius auratus) a small fish, while the dam water samples showed a population of larger fish. This difference between ecosystems affects the conductivity of the matrix that influences the peak shape, as already reported by Ollikainen and collaborators, who used the same instrumentation (HVStat) [69]. Using the nitrite peak area and the obtained linear regression from the quantification by the ME–AD, the nitrite concentration for the six samples was calculated. The results are presented in Table 3.

For all analyses, recoveries above 83.5% were achieved. Thus, the determination of nitrite by the ME–AD developed in this study is a promising methodology for analysis in environmental samples, with no detected matrix effect that changes the reliability of the determination. 

## 4. Conclusions

In this study, a promising methodology using ME–AD for the quantification of nitrite in water samples was developed. The use of a portable instrument has offered a satisfactory analytical performance. The proposed methodology was optimized, and the proof-of-concept was successfully demonstrated through the determination of nitrite in environmental samples of water from aquariums and dams. The obtained results revealed a good agreement with the data recorded by the reference technique. Based on the achieved results, the methodology developed in a portable and compact instrument may emerge as powerful analytical tool for in-field analysis.

## Figures and Tables

**Figure 1 micromachines-13-01736-f001:**
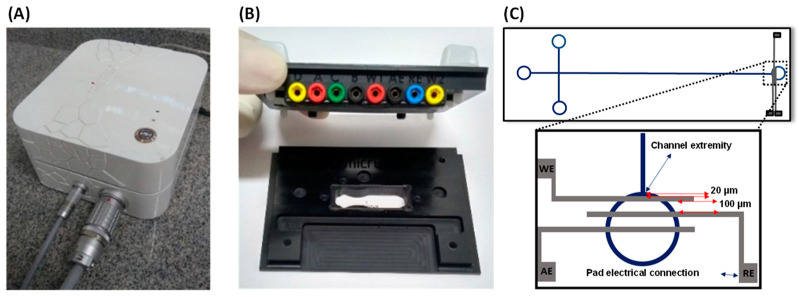
Portable instrumentation for ME–AD analysis including: (**A**) HVStat system manufactured by MicruX Technologies (Asturias, Spain) with integrated bipotentiostat coupled with a high voltage power source, (**B**) microfluidic holder (model MCE-HOLDER-DC02), (**C**) layout of the hybrid microchip composed of SU-8/glass with integrated electrodes for electrochemical detection highlighting the positioning of electrodes at end-channel arrangement.

**Figure 2 micromachines-13-01736-f002:**
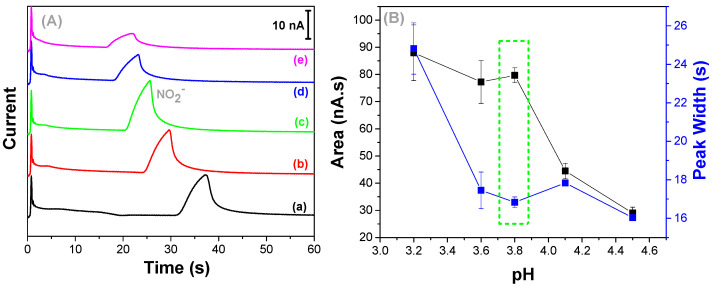
(**A**) Electropherograms showing the separation and detection of nitrite (50 µmol L^−1^) using a running buffer composed of 30 mmol L^−1^ lactic acid and histidine in the concentrations between 5 and 25 mmol L^−1^, resulting in a pH from 3.2 to 4.5 (a–e). (**B**) Peak area and width versus buffer pH values. Injection: −800 V for 10 s, Separation: −1000 V for 60 s, Detection 0.7 V versus Pt.

**Figure 3 micromachines-13-01736-f003:**
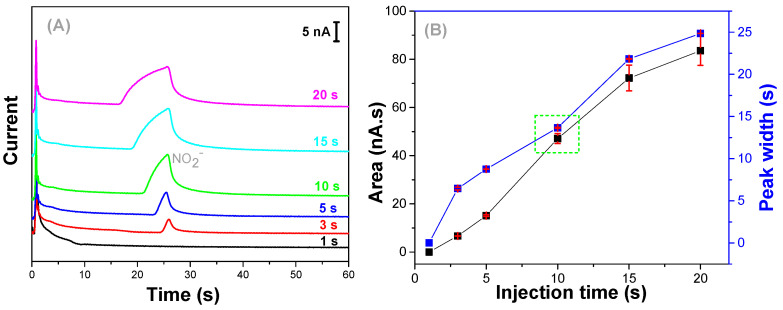
(**A**) Electropherograms showing the separation and detection of nitrite (50 µmol L^−1^) introduced into microchannels under different injection times: 1–20 s. (**B**) Area and peak width versus injection time. Running buffer: 30/15 mmol L^−1^ Lactic Acid/Histidine (pH = 3.8). Injection: −800 V, Separation: −1000 V for 60 s, Detection 0.7 V versus Pt.

**Figure 4 micromachines-13-01736-f004:**
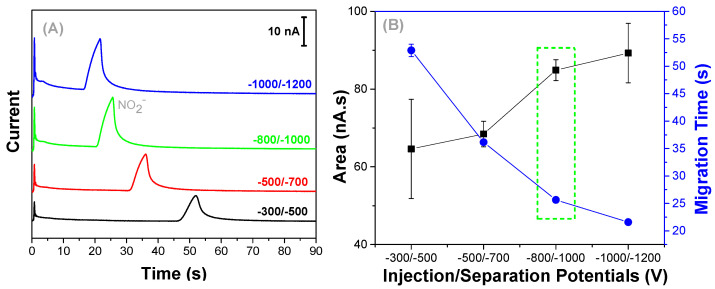
(**A**) Electropherograms showing the separation and detection of nitrite (50 µmol L^−1^) under different injection/separation voltages. (**B**) Peak area and migration time versus injection/separation potentials. Experimental conditions are given in Figure 3.

**Figure 5 micromachines-13-01736-f005:**
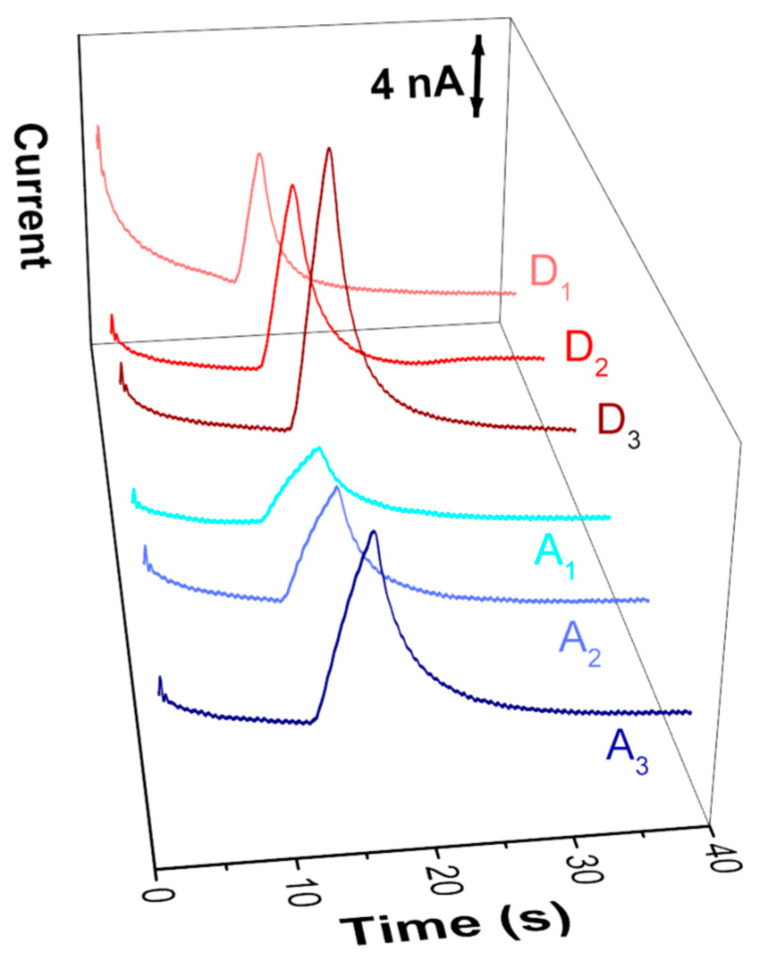
Electropherograms showing the detection of nitrite in (A1–A3) aquarium water samples (A1–A3) and fish breeding dam water samples (D1–D3). Injection: −800 V for 10 s; Separation: −1000 V for 60 s. For the other conditions, see Figure 3.

**Table 1 micromachines-13-01736-t001:** Summary of intra-day, inter-day, and inter-chip comparisons for the analysis of a 100 µmolL^−1^ nitrite standard solution (*n* = 3).

Parameter	Intra-Day	Inter-Day	Inter-Chip
RSD _Intensity_ (%)	0.1	12.3	11.8
RSD _Area_ (%)	2.3	15.6	9.8
RSD _Migration Time_ (%)	0.6	3.3	3.4

**Table 2 micromachines-13-01736-t002:** Comparison of the analytical performance of the proposed methodology for nitrite determination with other studies reported since 2017.

Sample	Analytical Technique	Analysis Time (s)	LOD (μmol L^−1^)	Portable	Ref
blood	IC—conductivity detection	~1920	0.078	No	[3]
water	HPLC—UV/DAD	180	9.78	No	[4]
food	CPE/UV–Vis	~900	0.003	No	[26]
water and food	Paper-based electrochemical devices	N/E	0.1	Yes	[38]
water	Paper-based colorimetric devices	~900	2.6	Yes	[17]
Saliva	Paper-based colorimetric devices	~300	4.8	Yes	[43]
Meat and water	Electrocatalysis/AD	~120	0.020	No	[41]
water	Electrocatalysis/AD	~3	0.038	No	[24]
water	Electrocatalysis/AD	~50	0.000184	No	[59]
water	ME-conductivity detection	~350 *	0.652	No	[44]
post-blast explosive residues	ME–C^4^D	150	9.5	Yes	[51]
water	ME–AD	80	2.8	No	[53]
water	ME–AD	70	8.2	No	[54]
Cells	ME–AD	35	0.50	No	[56]
water	ME–AD	50	1.3	Yes	This study

N/E: not specified. * Time of sample treatment not considered.

**Table 3 micromachines-13-01736-t003:** Nitrite concentration values found in water samples using the reference and the proposed methodologies (*n* = 3).

Sample	Reference Methodology (µmol L^−1^)	ME-DA (µmol L^−1^)	Recovery (%)	RSD (%)
D1	32.2 ± 0.3	30.4 ± 1.0	90.7–97.1	3.3
D2	52.2 ± 0.5	46.3 ± 5.3	88.8–98.6	6.3
D3	72.2 ± 0.6	67.1 ± 0.5	92.5–93.8	0.7
A1	30.0 ± 0.3	28.7 ± 1.6	90.1–100.8	5.6
A2	40.0 ± 0.4	35.9 ± 2.9	83.5–97.7	8.1
A3	60.0 ± 0.5	56.6 ± 5.0	87.8–103.8	8.8

## Data Availability

Not applicable.

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
