# Peer review of "Nitrite Determination in Environmental Water Samples Using Microchip Electrophoresis Coupled with Amperometric Detection"

_micromachines, 2022, doi:10.3390/mi13101736_

Round 1

Reviewer 1 Report

In this work, the authors developed a portable analytical methodology on chip-based electrophoresis coupled with amperometric detection for the determination of nitrite in environmental water samples. Such a method showed the satisfactory analytical performance for nitrite detection. It is recommended to be accepted for publication after a major revision, and the following issues should be addressed.

1)  Figure S1 should be move to the manuscript.

2)  The introduction section is miscellaneous, and the authors should reorganize this part to give a more concise introduction.

3)  Figure 2B is not mentioned in the manuscript, and the caption should be revised that the Y axis is “Intensity and Migration Time” rather than “Area and Migration Time” in the current version.

4)  There are many typos in the manuscript, please carefully check the text.

Reviewer 2 Report

Title: Nitrite determination in environmental water samples using microchip electrophoresis coupled with amperometric detection

 major revision need

 1. In this work, pH used 3.8. Its acid ranges. what is the major reason for this? 

2.  Authors are mentioned "microchip electrophoresis coupled with amperometric detection, But, I can’t see amperometric detection data in this manuscript. 

3. Introduction section must be revise more effectively and follow more suitable research articles. Follow those articles for nitrite detection (https://www.sciencedirect.com/science/article/pii/S0013468621005703, https://www.nature.com/articles/s41598-017-12050-x, https://www.sciencedirect.com/science/article/pii/S001393512200072X#!

https://link.springer.com/article/10.1007/s00604-017-2379-9,

https://www.sciencedirect.com/science/article/pii/S0026265X21009905

https://link.springer.com/article/10.1007/s00604-022-05296-4)

4. Results: Some experimental data lack (relative) standard deviations. Give averaged data for important experimental data along with standard deviations (±) and the number of experiments (n =?).

5. Figure quality need to improve.

6. There are few grammatical and typographical errors. Please check the manuscript and refine carefully.

Reviewer 3 Report

The authors developed a microchip electrophoretic method coupled with amperometric detection for analysis of nitrite in environmental water samples. The method was validated, and the validation studies proved linearity, precision, and accuracy of the developed method. Although, the literature revealed different microchip electrophoretic methods for the analysis of nitrite in biological and environmental matrices, this report deserves publication in micromachines after revision because it involved simple, fast, precise, and selective analytical method for the assay of nitrite.

1.     The authors are asked to compare between the developed method and the recently developed methods for nitrite sensing in terms of sensitivity, selectivity, practicability, rapidity, and stability to illustrate why this developed method is superior to the previously developed ones. This comparison could be performed in a tabular form.

2.     Some references that were related to analysis of nitrite by microchip electrophoresis are missed. For instance: 

a.     Siegel, J.M., Schilly, K.M., Wijesinghe, M.B. et al. (2019). Optimization of a microchip electrophoresis method with electrochemical detection for the determination of nitrite in macrophage cells as an indicator of nitric oxide production. Anal. Methods 11, 148–156. https://doi.org/10.1039/C8AY02014K.

b.     Troška, P., Chudoba, R., Danč, L., Bodor, R., Horčičiak, M., Tesařová, E., & Masár, M. (2013). Determination of nitrite and nitrate in cerebrospinal fluid by microchip electrophoresis with microsolid phase extraction pre-treatment. J. Chromatogr. B930, 41–47. https://doi.org/10.1016/j.jchromb.2013.04.042

c.      Moravský, L, Troška, P, Klas, M, Masár, M, Matejčík, Š (2020). Determination of nitrites and nitrates in plasma-activated deionized water by microchip capillary electrophoresis. Contributions to Plasma Physics. 60: e202000014https://doi.org/10.1002/ctpp.202000014.

d.     Kikura-Hanajiri, R., Martin, R. S., & Lunte, S. M. (2002). Indirect measurement of nitric oxide production by monitoring nitrate and nitrite using microchip electrophoresis with electrochemical detection. Analytical chemistry, 74, 6370–6377. https://doi.org/10.1021/ac0204000.

Please cite the missed references properly in the text.

3.     The manuscript should also be revised carefully for language and grammatical errors because it contains some linguistic and grammatical mistakes.

Round 2

Reviewer 1 Report

The manuscript was significantly improved after the revision, and it was recommended to be accepted for publication.